# Usefulness of Docking and Molecular Dynamics in Selecting Tumor Neoantigens to Design Personalized Cancer Vaccines: A Proof of Concept

**DOI:** 10.3390/vaccines11071174

**Published:** 2023-06-29

**Authors:** Diego Amaya-Ramirez, Laura Camila Martinez-Enriquez, Carlos Parra-López

**Affiliations:** 1INRIA Nancy Grand Est, Université de Lorraine, 54600 Nancy, France; diego.amaya-ramirez@univ-lorraine.fr; 2Grupo de Inmunología y Medicina Traslacional, Departamento de Microbiología, Facultad de Medicina, Universidad Nacional de Colombia, Bogotá 111321, Colombia; lamartineze@unal.edu.co

**Keywords:** molecular docking, molecular dynamics, personalized cancer vaccines, cancer immunotherapy

## Abstract

Personalized cancer vaccines based on neoantigens are a new and promising treatment for cancer; however, there are still multiple unresolved challenges to using this type of immunotherapy. Among these, the effective identification of immunogenic neoantigens stands out, since the in silico tools used generate a significant portion of false positives. Inclusion of molecular simulation techniques can refine the results these tools produce. In this work, we explored docking and molecular dynamics to study the association between the stability of peptide–HLA complexes and their immunogenicity, using as a proof of concept two HLA-A2-restricted neoantigens that were already evaluated in vitro. The results obtained were in accordance with the in vitro immunogenicity, since the immunogenic neoantigen ASTN1 remained bound at both ends to the HLA-A2 molecule. Additionally, molecular dynamic simulation suggests that position 1 of the peptide has a more relevant role in stabilizing the N-terminus than previously proposed. Likewise, the mutations may have a “delocalized” effect on the peptide–HLA interaction, which means that the mutated amino acid influences the intensity of the interactions of distant amino acids of the peptide with the HLA. These findings allow us to propose the inclusion of molecular simulation techniques to improve the identification of neoantigens for cancer vaccines.

## 1. Introduction

Personalized neoantigen-based vaccines have proven to be a useful tool for immunotherapy of aggressive tumors, such as metastatic melanoma, glioblastoma, and non-small-cell lung cancer [1,2,3,4,5,6]. This type of vaccine promotes a highly specific response of T lymphocytes that do not undergo the tolerance induction process in the thymus because they recognize neoantigens encoded in the somatic mutations of the tumor [7,8,9].

Currently, tumor neoantigens are predicted using in silico strategies, which compare the tumor DNA sequence with that of healthy patient tissue to identify tumor-specific somatic mutations. Once the mutations have been identified, the neoantigens are predicted by using trained algorithms to estimate the probability that they will be processed and presented in the context of the major histocompatibility complex (MHC) molecules, also called human leukocyte antigen (HLA), of the patient [10,11,12,13,14,15]. Once the identity of the neoantigens is established, the sequence of the tumor transcriptome allows the expression levels of the predicted neoantigens to be verified. Other parameters incorporated in bioinformatic tools for the identification of immunogenic neoantigens are the relative affinity of the MHC molecule–neoantigen complex (IC50) [16,17,18] and the half-life of the binding of the MHC–neoantigen complex [19].

These in silico strategies allow the prioritization of potentially immunogenic neoantigens. However, despite the promising results of some studies on the design and clinical response of tumors to personalized vaccines, clinical studies have shown a limited in vivo immunogenicity of the selected neoantigens [20], likely due to a limited capacity of in silico methods to identify immunogenic epitopes in vivo. Despite the progress of in silico prediction tools, they still have a high rate of false positives (low specificity) [21,22], which is mainly due to two factors: (1) the dataset used to train these tools is usually based on information generated from a limited number of HLA alleles, and they do not take into account results of in vitro or in vivo evaluation of neoantigens; and (2) predictive tools rely on sequence data only and therefore do not satisfactorily incorporate molecular aspects of epitope processing and presentation by antigen-presenting cells, such as stability of the MHC–peptide complex and recognition of the MHC–peptide complex by the T cell receptor (TCR) on T cells, which are important factors determining the immunogenicity of antigens [23].Therefore, it is necessary to search for new tools to improve the selection of peptides with immunogenic potential in vivo.

Docking and molecular dynamics are computational tools that allow the understanding of non-covalent receptor–ligand interactions at an atomic level useful in rational drug design [24] and in the study of the interaction of the HLA–peptide complex with the TCR molecules [25,26,27,28,29]. The use of these tools in the selection of immunogenic tumor neoantigens have not been explored. Docking and molecular dynamics might make it possible to generate additional information on the interaction of HLA molecules with neoantigens and, perhaps, to obtain a better selection of neoantigens efficiently presented by MHC molecules and recognized by T cells in vivo.

In this work, molecular docking and molecular dynamics were used to discriminate molecular interactions among HLA-A*02:01 molecules and two neoantigens, one immunogenic and one non-immunogenic, for T cells. Based on the study by Strønen et al., [30], we chose two melanoma neoantigens that, according to predictive algorithms, could bind to HLA-A*02:01 molecules with high affinity and whose data on immunogenicity were available. To elucidate the structural properties of a neoantigen in a complex with HLA-A*02:01 leading the expansion of the human cytotoxic CD8+ T cell that efficiently recognizes and destroys melanoma cells, we analyzed both complexes through the aforementioned techniques and found remarkably different structural features on each complex.

## 2. Materials and Methods

### 2.1. Neoantigen Selection

Based on the study published by Strønen et al. [30], 21 neoantigens and their wild-type counterparts were selected (see Table 1). The peptides had the following characteristics: (i) the amino acid sequence of the mutated peptide included a single amino acid change, compared to the wild-type sequence; (ii) the affinity score of the mutant peptide was <1000 nM (according to NetMHC 3.2); (iii) the stabilities of the peptide–HLA-A*02:01 complexes were measured; and (iv) the immunogenicity for CD8+ T cells were assessed experimentally.

The peptide–HLA-A*02:01 complexes of these 42 peptides were modeled by molecular docking, and the number of total and hydrophobic interactions were analyzed with LigPlot+ [31]. Due to the elevated computational cost of molecular dynamics, only two neoantigens and their wild-type counterparts were selected for this simulation. The selection was based on the number of total interactions between the peptide and HLA (Table 2) and the immunogenicity reported by Strønen (Table 1). Considering these criteria, the wild-type and neoantigen associated with the gene ASTN1 (10 amino acids in length) and AKAP6 (9 amino acids in length) were chosen as immunogenic and non-immunogenic neoantigens, respectively.

### 2.2. Evaluation of Peptides through Sequence-Based In Silico Tools

The NetMHC 4.0 tool for HLA-A*02:01 binding affinity assessment [32], NetMHCstabpan 1.0 to predict MHC-I binding stability (19), NetCTL 1.2 and NetTepi 1.0 to determine proteasome processing, TAP transport, and HLA binding [33,34] were all used to predict the immunogenicity of the CD8+ T cells of the four peptide sequences listed in Table 3.

### 2.3. Molecular Docking

To generate the peptide–HLA complex model for each case, molecular docking was performed. To model the peptide pair derived from ASTN1 (wild-type and mutant neoantigen), the structure 5C0G from the Protein Data Bank (PDB) [35] was used as a reference structure. For the pair of peptides derived from AKAP6, the 5NMH structure was used. The FlexPepDock Ab-Initio protocol [36] integrated in the Rosetta package [37] was implemented for docking, where 50,000 poses were calculated for each peptide. According to the Rosetta scoring function, the best pose generated for each peptide was selected. The number of interactions (hydrogen bonds and hydrophobic interactions) of the peptide–MHC complex were evaluated through Ligplot+ v2.2.

### 2.4. Molecular Dynamics Simulation

Subsequently, a molecular dynamics simulation was performed for each peptide using the Visual Molecular Dynamics (VMD) [38] and Nanoscale Molecular Dynamics (NAMD) [39] tools. For the generation of the topology of the system, the VMD AutoPSF tool [38] was used together with the CHARMM36 force field [40], the system was solvated using the TIP3P explicit solvent model, the size of the water box was 67 × 70 × 70 A3, the system was neutralized using the VMD AutoIonize tool [38], and the particle mesh Ewald (PME) method was used to calculate the electrostatic energy, with a distance truncation of 11 Å. The simulation was carried out under NPT conditions, that is, constant pressure (1 atm) and temperature (310 K), and was composed of 3 stages: (i) minimization, for which the system was brought to room temperature (310K), (ii) system stabilization for ~15 ns, and (iii) the simulation itself, with a duration of 295ns. Lastly, the VMD [38] and Pycontact [41] tools were used to analyze both the peptide–protein interactions and to assess the complex stability.

## 3. Results

Initially, molecular docking was carried out to model the peptide–MHC complexes (p-MHC) of the 21 neoantigens listed in Table 1 and their wild-type counterparts. The best pose of each of the peptides on HLA-A*02:01 was analyzed using the Ligplot+ tool to determine the number of hydrophobic interactions and hydrogen bonds in the p–MHC complex for both the mutant version (Table 2) and the wild-type version (Appendix A). Overall, the total interactions ranged from 16–29 without observing a pattern that allowed us to correlate immunogenicity with the number of interactions. Due to the elevated computational, we proceeded to select only two neoantigens, one immunogenic and the other non-immunogenic, to perform the simulation of the p–MHC complex through molecular dynamics to analyze the peptide–protein interactions and the stability of the complex over time. Therefore, based on the number of interactions by molecular docking and the immunogenicity reported by Strønen, the AKAP6 peptide with 29 total interactions was selected as the non-immunogenic neoantigen, and ASTN1 with 26 total interactions was selected as the immunogenic neoantigen to continue with the molecular simulations.

### 3.1. Limitation of the Predictive Tools Based on Sequence

The AKAP6 neoantigen was generated by substituting glutamic acid for a lysine (E/K) at position 6 (P6). The predictive analysis of the AKPK6 neoantigen sequence (IC50 and complex stability values) suggested that both the wild-type and the mutant sequences should be more immunogenic than the ASTN1 neoantigen for CD8+ T cells; however, the in vitro evaluation carried out by Strønen proved that neither the wild-type nor the mutant sequence of AKAP6 were immunogenic. On the other hand, the ASTN1 neoantigen was generated by changing a proline for a leucine (P/L) at position 2 (P2), whereas the NetCTL tool (which predicts cleavage by the proteasome and efficiency of transport by TAP) revealed that ASTN1 did not meet the characteristics of an immunogenic sequence. The NetMHC 4.0 and NetMHCstabpan 1.0 tools revealed that ASTN1 neoantigen had a higher affinity for the HLA-A*02:01 molecule and formed stable MHC–peptide complexes, predicting a more immunogenic sequence than that formed by the wild-type sequence. This was proved experimentally by Strønen, since the in vitro evaluation of this neoantigen clearly demonstrated that this neoantigen was highly immunogenic for CD8-T cells (Table 3). Altogether, our results suggest that predictive algorithms provide conflicting results that are hard to conceal with epitope immunogenicity and argue for the need to have other techniques to improve the prediction of immunogenic epitopes.

### 3.2. Molecular Simulations

From the structures derived from molecular docking for AKAP6 and ASTN1, both for the wild-type versions and for the neoantigens, a peptide–HLA binding was observed, with a conventional orientation with the side chains of the P2 and P9/P10 residues arranged in the pockets of HLA-A2 (Figure 1A,B). For AKAP6, P2 was leucine, and P9 was valine; for ASTN1, P2 was proline/leucine (WT/Neo), and P10 was leucine. The exposed side chains for AKAP6 were P1, P5, and P8, and for ASTN1, they were P1, P5, P6, P8, and P9, all projected away from the HLA-A2 binding pockets, forming a surface with the potential to interact with HLA-A2.

In both cases, the peptide–MHC complexes showed structural deviations between the wild-type version and the neoantigen (Figure 1C,D). In particular, the wild-type and mutant peptides of AKAP6 overlapped in all the side chains except for P1, P3, and P6, which were the last ones where the mutation occurred. For its part, in ASTN1, only P2 and P3 changed, being the first where the mutation occurred. Therefore, the structural differences between the peptide–MHC complexes for the wild-type and the mutant versions in both neoantigens appeared not to be restricted solely to the mutation site.

The molecular dynamics simulations showed very different structural attributes of both neoantigens bound to the HLA-A*0201 molecule. On one hand, for AKAP6, both the wild-type (link to visualize online the simulation: https://mmb.irbbarcelona.org/3dRS/shared/61b278eec0a8d2.72605506 (accessed on 10 December 2021), corresponding files are also available at Zenodo with https://doi.org/10.5281/zenodo.5772726 (accessed on 10 December 2021)) and the mutant versions (link to visualize online the simulation: https://mmb.irbbarcelona.org/3dRS/shared/61b27a3cdfdef2.66763559 (accessed on 10 December 2021), corresponding files are also available at Zenodo with https://doi.org/10.5281/zenodo.5772726 (accessed on 10 December 2021)) bound to the HLA-A*02:01 molecule; however, by the end of the simulation, the C-terminus end of both peptides dissociated from the peptide binding groove (PBG) of the HLA-A*02:01 molecule, remaining anchored only by the N-terminus (P1 to P5). In the case of ASTN1, the simulations showed that the wild-type version of the peptide (link to visualize online the simulation: https://mmb.irbbarcelona.org/3dRS/shared/61b27dac052904.99817841 (accessed on 10 December 2021), corresponding files are also available at Zenodo with https://doi.org/10.5281/zenodo.5772726 (accessed on 10 December 2021)) detached from the N-terminus end and remained anchored to the MHC molecule through the C-terminus end (P7 to P10). In contrast, the neoantigen sequence of ASTN1 remained anchored at both ends of the MHCI throughout the simulation time, meaning that the amino acid change in P2 allowed the ASTN1 neoantigen to form the most stable peptide–HLA-A*02:01 complex of the four MHCI–peptide complexes analyzed (link to visualize online the simulation: https://mmb.irbbarcelona.org/3dRS/shared/61b27c62dc9300.82271714 (accessed on 10 December 2021), corresponding files are also available at Zenodo with https://doi.org/10.5281/zenodo.5772726 (accessed on 10 December 2021)).

### 3.3. Atomic Interactions

Atomic interactions between the peptide and HLA-A*02:01 were analyzed using the Pycontact tool [41]. Pycontact is a bioinformatics tool that identifies and characterizes non-covalent interactions between molecules in a molecular dynamic simulation. More specifically, it calculates the intensity of interactions through a magnitude called “contact score.” The stronger the interaction between two atoms/residues, the higher the “contact score.” This analysis focused on stable interactions over time, that is, those interactions with a median “contact score” greater than zero. At first, the number and type of stable interactions were analyzed, with the predominant types of interactions being hydrophobic and hydrogen bonds (Figure 2). Regarding the types of interactions considered here, it is worth clarifying that the “other” category corresponds to the interactions that did not strictly meet the classification thresholds of any of the other categories.

In the case of AKAP6, even though both the wild-type peptide and the neoantigen detached from the C-terminus end (more precisely, positions P6 to P9), the mutated peptide presented a reduction in the number of interactions in the C-terminus part, generating faster release, compared to the wild-type version (Figure 2A). In the case of ASTN1, the mutation in this neoantigen generated an increase in the number of stable interactions with HLA, not only at the position where the amino acid change occurred (P2), but also at other positions (notably at the P9). This increase in interactions favored the stability of the complex, as evidenced in the molecular simulation, since it was this peptide that remained anchored at both ends, contrary to the wild-type, which was released from the N-terminus end.

On the other hand, the variability of the types of interactions between neoantigens and their wild-types were striking. This illustrates the complexity of the dynamics of protein–peptide interactions and helps to understand why current tools trained primarily on sequence information are unable to accurately predict the stability of peptide–HLA complexes.

### 3.4. Intensity of Interactions (Contact Score)

The intensity of interactions between atoms/residues is of great interest when it comes to characterizing the stability of a complex. In AKAP6, a significant increase in the average intensity of interaction in P3 and P4 between the wild-type and the mutated versions was observed (Figure 3). However, this change was irrelevant in the global stabilization of the peptide, since, in both cases, the peptide was released from the C-terminus end.

As for ASTN1, the mutation in P2 generated increases in the intensity of the interaction not only at the site of the mutation but also at other positions, such as P1, P7, and P9 (see Figure 3). This result is particularly interesting because it indicates a “delocalized” (long-distance) impact of a point mutation on the global peptide–HLA interaction. In other words, a point mutation can modulate the interactions of other peptide amino acids with the PBG of the HLA molecule.

### 3.5. Atomic Interactions of P1 of AST1 with the HLA-A*02:01 Molecule

We focused the docking and molecular dynamics analyses on the atomic interactions between residues of P1 with the peptide binding groove of HLA-A*02:01 in ASTN1, particularly in the ten strongest atomic interactions (Table 4).

Interactions of P1 with the HLA-A*02:01 molecule in the neoantigen were not found in the wild-type version of the sequence interacting with this molecule. (Where three out of four interactions were hydrogen bonds, the rest corresponded to a different type of non-covalent interaction; see Figure 4). This observation is interesting, since it allowed us to measure the magnitude of the “delocalization effect” of the mutation in P2 on other positions along the peptide. By this effect, P1 became the position that fostered the most intense interactions stabilizing the peptide.

### 3.6. Atomic Interactions between P2 and Its Neighbors in ASTN1

Due to the mutation, the intensity of the interactions of P1 with the HLA-A*02:01 changed considerably between the ASTN1 neoantigen and the wild-type sequence (Figure 3). Therefore, we investigated the effect of the mutated position on neighboring amino acids, and we found that in the wild-type peptide, the proline in P2 established a very strong interaction with P1 that resulted in an accumulated contact score that reached a value of 19.14 against 13.33 in the case of the neoantigen (Appendix A). This could explain the interactions observed at P1 of the ASTN1 neoantigen (Table 4) that were lost between the P1 of the wild-type peptide and the HLA-A*02:01 molecules, which would be hindered by proline. Regarding the interactions between P2 and P3, it is worth noting that no significant changes were found in the accumulated contact score (12.67 in the case of the wild type and 11.72 in the case of the neoantigen; see Appendix A).

These findings highlighted new features governing the complexity of peptide–HLA interactions that might explain why current bioinformatics tools are unable to accurately predict the affinity and stability of this type of complex.

### 3.7. Analyses of Immunogenic vs. Non-Immunogenic Interactions

When comparing the non-immunogenic with the immunogenic neoantigens (AKAP6 vs. ASTN1, respectively), it was noticed that not only the amount and intensity of the interactions mattered, both ends (N-terminus and C-terminus) remained stably bound to the HLA’s PBG throughout the time. Also noteworthy was the fact that in the case of both AKAP6 peptides (wild-type and mutant) and in the ASTN1 neoantigen, which are the peptides that remained anchored in the N-terminus part, P1 had both a significant number of interactions and a high intensity (even higher than P2, Figure 3), which would indicate that this position played a more relevant role than traditionally accepted in the stabilization of the N-terminus part of peptide epitopes.

## 4. Discussion

In the present work, we explored the use of molecular simulation techniques, such as docking and molecular dynamics, to study the interaction and stability of peptide–HLA complexes. For this, AKAP6 was selected as the non-immunogenic neoantigen and ASTN1 as the immunogenic antigen, according to the results obtained in healthy donors for Strønen et al. [30], to use these tools in silico and analyze the characteristics that define their immunogenicity at the pMHC complex level. When comparing the in vitro immunogenicity results with the results obtained by traditional prediction tools, the failures of algorithms based solely on the sequence were evident, since AKAP6 was a clear false positive. These results can be explained by the lack of structural information on the interaction of the p–MHC complex that these tools have. Therefore, the inclusion of docking and molecular dynamics may help to strengthen the prediction of immunogenic neoantigens.

Regarding the two cases selected for this work, the results provided by molecular dynamics highlighted three characteristics when comparing these two neoantigens: (i) the importance of P1 and P2 for the binding of the peptide and the MHC; (ii) the delocalized effect that mutations can have and how this can influence the stability of the peptides; and (iii) the importance of having high affinity and stability of the complex at both ends.

First, the results provided by the molecular dynamics of this study indicated that the P1 position of the peptide played more of a key role in the stabilization of the peptide by the N-terminus than previously assumed, since classically, P2 and P9 were the residues of the most important peptides for anchoring to the MHC [42]. This prominent role of P1 is evident in Figure 2, where significantly more important contact scores were observed in P1 than in P2 in the three peptides that remained anchored in their N-terminus part (ASTN1 neo, AKAP6 WT, and AKAP6 neo). The two neoantigens followed the classically reported amino acid sequence pattern for class I epitopes for binding to MHC-I: X-(L/I)-X(6−7)-(V/L), where L/I and V/L represent the residues, whose side chain anchors the peptide to the MHC [42,43]. In the case of AKAP6, this was generated by a mutation in a non-anchor position (P6), which caused the mutant peptide to be released more quickly from the MHC through the C-terminus. On the other hand, in ASTN1, the mutation occurred right at an anchor position (P2) with a change from a proline to a leucine, which improved the anchoring of the peptide to the MHC and increased the number of hydrophobic interactions. These two characteristics, mutations in the anchoring residues (P2 and P9) that improve the affinity and increase the hydrophobicity of the amino acids, were reported in literature as properties related to immunogenicity, since stable interactions are generated between the anchoring amino acids and the HLA, which allow the correct presentation of the antigen [44,45].

Second, the “delocalized” impact of a point mutation on the overall peptide–HLA interaction was evidenced, meaning that a mutation could modulate the interactions of other peptide amino acids with HLA (see Table 4). This kind of effect was previously shown in the context of interaction with the TCR, since certain mutations induce structural changes in the amino acids involved in recognition by the TCR, due to movement fluctuations that the side chains may have, due to the mutation [46,47,48]. This means that substitutions in a peptide can alter the intra-residual interactions, which can potentially alter its conformation and, therefore, its recognition by the TCR.

Finally, the stability results of the pMHC complexes obtained agreed with the in vitro immunogenicity results, which would indicate the utility of this type of in silico strategy in identifying peptides that form stable complexes with HLA proteins. When comparing the two neoantigens, it was possible to show that the immunogenic peptide (ASTN1) was linked to both ends (N-terminus and C-terminus) in a stable manner, over time, on HLA. These results support the importance of stability, in addition to binding affinity, as a key factor for the selection of immunogenic neoantigens [30,33,49], since it plays a key role in the adequate presentation of the peptide to the LT and thus triggers an immune response. This is reinforced by previous studies that reported failures in the selection of neoantigens when the main or only parameter considered was the binding affinity [23,50,51,52].

Even though only two peptides were analyzed, the results shed light on characteristics of immunogenicity; however, these must be validated in larger cohorts to define the role they have in binding affinity and stability in the MHC and, finally, on immunogenicity. However, due to the considerable computational cost of this type of strategy, its use would be restricted to the final stages of the immunogenic neoantigen identification pipelines, when a small number of candidates remain. These tools have the potential to work not only at the level of the interaction within the peptide and the MCH but also to determine the interactions between the p–MHC complex and TCR, which can eventually be implemented to select the best TCRs for adoptive therapy purposes.

## 5. Conclusions

Personalized cancer vaccines are presented as a novel and promising alternative to cancer immunotherapy, especially in cases where effective treatments do not yet exist. Currently, most selections of neoantigens are made by in silico methodologies [15]; however, the results of clinical studies reveal that the in vivo immunogenicity of in silico epitopes is very limited. Therefore, it is important to search for new tools that allow the selection of immunogenic epitopes that yield better clinical results in vivo when used as a vaccine. An important attribute of immunogenic MHC–peptide complexes is the long half-life these complexes have [33,49,53,54,55]; therefore, we believe that molecular simulation can play an important role in the fine-tuning of the selection process required to select neoantigens to be included in a neoantigen vaccine. In the present work, we explored the use of molecular simulation techniques, such as docking and molecular dynamics, to analyze the role of peptide–HLA complex stability in immunogenicity for CD8+ T lymphocytes. The analysis of the stability of the analyzed complexes collated with the results of immunogenicity in vitro confirmed that kind of relationship. These results point to the suitability of this type of in silico strategy to identify peptides that form stable complexes with HLA proteins that are highly immunogenic for CD8+ T cells.

## Figures and Tables

**Figure 1 vaccines-11-01174-f001:**
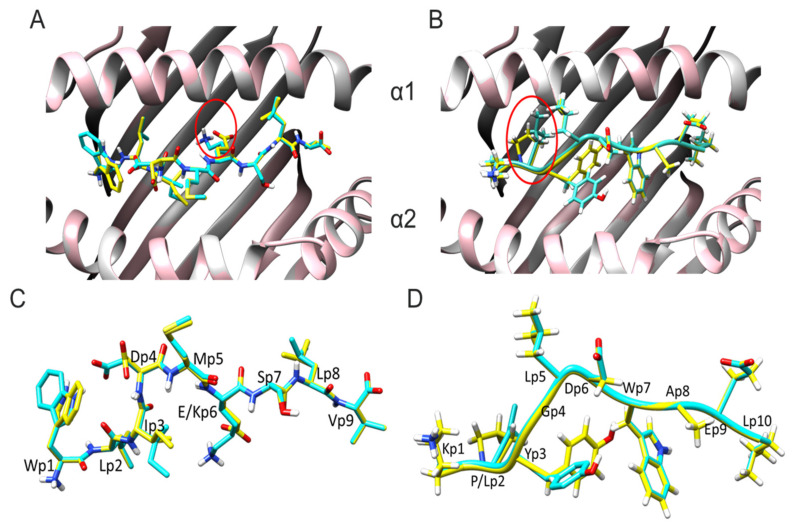
Conformers of wild-type and mutant peptides for AKPK6 and ASTN1 bound to HLA-A*02:01. (**A**) Top view of the AKAP6wt–HLA-A2 and AKAP6 neo-A2 complexes. (**B**) Top view of the ASTN1wt–HLA-A2 and ASTN1–neo-A2 complexes. The wild version is in yellow, and the mutant is in cyan. The point mutation is marked in a red circle. The HLA-A2 backbone is in white (WT-HLA-A2) or pink (neo-HLA-A2). (**C**). Side view of overlapping AKAP6 mutant and wild-type peptides. (**D**) Side view of overlapping wild-type and mutant ASTN1 peptides. Carbon atoms are in yellow (WT) or in cyan (Neo); nitrogen atoms are in blue; oxygen atoms are in red.

**Figure 2 vaccines-11-01174-f002:**
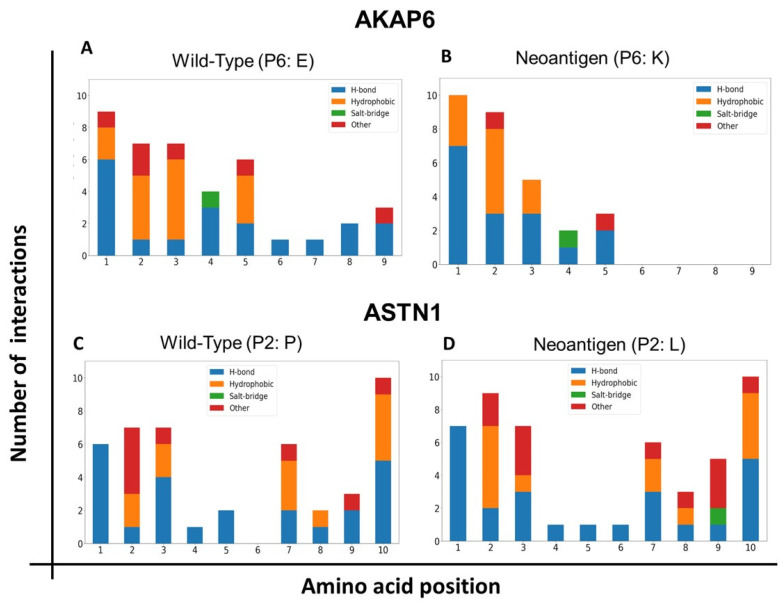
Number and type of interactions between each of the amino acids of the antigens and HLA-A*02:01. Molecular dynamics simulations were performed for the wild-type and the mutant version for both neoantigens, by analyzing the number and type of non-covalent interactions between the amino acids of the epitopes and HLA-A2 using the Pycontact tool. (**A**,**B**) Bar graphs for the non-immunogenic neoantigen AKAP6. (**C**,**D**) Bar graphs for the immunogenic neoantigen ASTN1. In blue are hydrogen bonds, in orange are hydrophobic interactions, in green are salt bridges, and in red are other types of interactions.

**Figure 3 vaccines-11-01174-f003:**
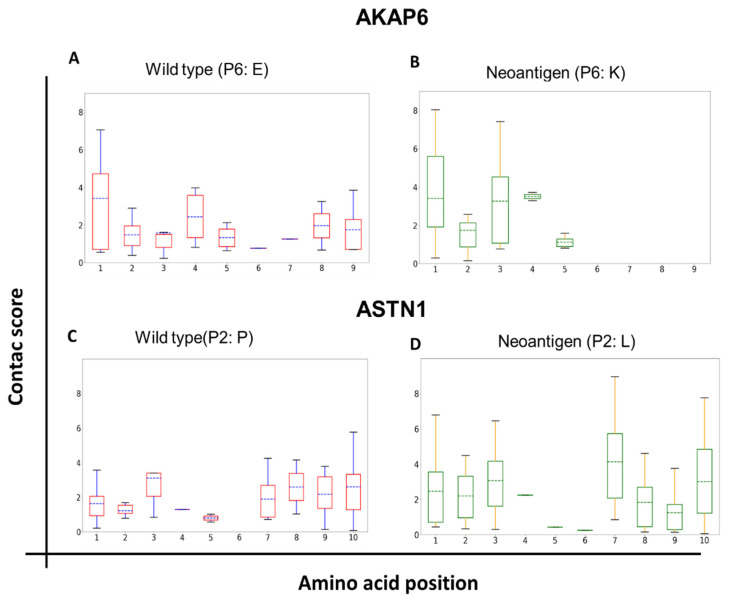
Contact scores of the interactions between amino acids of the antigens and HLA-A*02:01. Molecular dynamics simulations were performed for the wild-type and the mutant version for both neoantigens, by analyzing the intensity of the non-covalent interactions between the amino acids of the epitopes and HLA-A2. (**A**,**B**) Box-and-whisker plots for the non-immunogenic neoantigen AKAP6. (**C**,**D**) Box-and-whisker plots for the immunogenic neoantigen ASTN1.

**Figure 4 vaccines-11-01174-f004:**
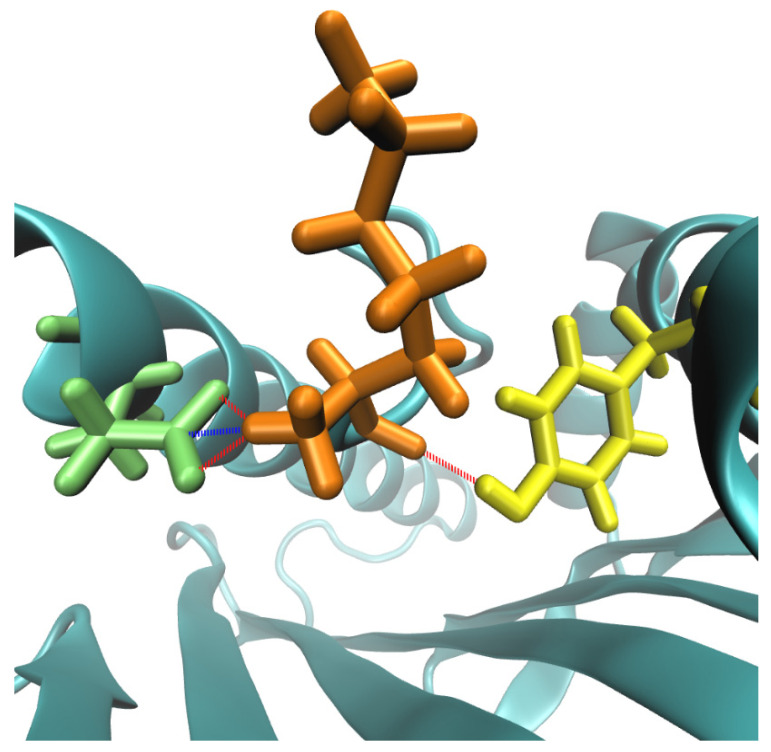
Visualization of the four interactions differentially present in the ASTN1 mutant peptide, compared to its wild-type counterpart. The first amino acid of the neoantigen ASTN1 (K) is shown in orange, and the amino acids 159 and 63 of the HLA are in yellow and green, respectively. Hydrogen bonds are shown in red, and another type of interaction is shown in blue.

**Table 1 vaccines-11-01174-t001:** Peptide sequences of the wild-type and mutant neoantigens selected for assessment using molecular docking amino acid substitutions in the mutant neoantigens are presented in red.

Gene Associated	Wild-Type Sequence	Mutant Sequence	Neoantigen Predicted Binding Affinity (nM)	Neoantigen t1/2 β2 Microglobulin (Hours)	Immunogenicity: CD8+ T Cell Response ^1^
AKAP6	WLIDMESLV	WLIDMKSLV	24	12.3	NO
ASTN1	KPYGLDWAEL	KLYGLDWAEL	43	8.4	YES
BCSIL	ALALARKGV	ALALAQKGV	925	9.2	NO
BCSIL	ALARKGVQL	ALAQKGVQL	914	4.4	NO
CDK4	ARDPHSGHFV	ALDPHSGHFV	119	47.5	YES
DC1	VMKFKNPPV	VMKFKNPLV	350	3.4	NO
GCNIL1	ALLETLSLLL	ALLETPSLLL	27	39.9	YES
GNL3L	NLNRCSVPV	NLNCCSVPV	17	23.5	YES
GOLGA3	SLDPTTSPV	SLDLTTSPV	33	13.7	NO
HELLS	VTNSGKFLI	VTYSGKFLI	888	3.7	NO
KIF3B	SALGNVISA	FALGNVISA	238	4.6	NO
LAMA1	STASDFLAV	STAFDFLAV	363	3.7	NO
MLL2	ALSPVIPLI	ALSPVIPHI	18	47.7	YES
MRM1	LLFGMTPCL	LLFGMPPCL	37	14	YES
PGM5	AVGSHVYSV	AVGSYVYSV	15.8	12.4	YES
PGM5	QQFAVGSHV	QQFAVGSYV	73.53	5.4	YES
SIVA1	ALCGQCVRT	ALCGQCVRI	633	5	NO
SLC38A1	IWAALFLGL	ILAALFLGL	18	6	YES
SMARCD3	KLFEFLVHGV	KLFEFLVYGV	6	83.7	YES
SNX24	KLSHQPVLL	KLSHQLVLL	42	24	YES
USP28	LIIPCIHLI	LIIPFIHLI	28	10	YES

^1^ CD8 response was evaluated by multimers, and subsequent reactivity was measured by the expression of CD107 A/B and IFN-y of tetramer-positive clones against tumor cells in vitro [30].

**Table 2 vaccines-11-01174-t002:** Number of total interactions and hydrophobic interactions of mutant peptides with the corresponding HLA molecule.

Peptide	Sequence	Total Interactions	Hydrophobic Interactions
AKAP6	WLIDMKSLV	29	18
DC1	VMKFKNPLV	28	18
ASTN1	KLYGLDWAEL	26	16
GCNIL1	ALLETPSLLL	25	14
GNL3L	NLNCCSVPV	25	14
HELLS	VTYSGKFLI	24	15
SMARCD3	KLFEFLVYGV	24	13
USP28	LIIPFIHLI	24	15
GOLGA3	SLDLTTSPV	23	13
MRM1	LLFGMPPCL	23	15
PGM5	QQFAVGSYV	23	11
MLL2	ALSPVIPHI	22	13
SIVA1	ALCGQCVRI	21	11
SLC38A1	ILAALFLGL	21	13
BCSIL	ALAQKGVQL	20	11
CDK4	ALDPHSGHFV	19	10
LAMA1	STAFDFLAV	19	9
SNX24	KLSHQLVLL	19	10
BCSIL	ALALAQKGV	18	11
KIF3B	FALGNVISA	18	9
PGM5	AVGSYVYSV	16	8

**Table 3 vaccines-11-01174-t003:** Results of analyses of four peptides assessed by using the different tools traditionally used for the identification of neoantigens.

Associated Gene	Gene Function	Mutation	Peptide Sequence	In Silico				In Vitro		
				NetMHC 4.0 (Afinity: nM)	NetTepi 1.0 (Epitope)	NetCTL 1.2 (Epitope)	NetMHCstabpan 1.0 (Stability: hours)	t1/2 β2 Microglobulin (Stability: hours)	CD8+ T Cell Response Observed in Patient ^1^	CD8+ T Cell Response Induced in Donor ^1^
ASTN1	Neuronal adhesion molecule required for migration of neuroblasts	WT	KPYGLDWAEL	12,901	NO	NO	0.26	-	NO	NO
		P/L	KLYGLDWAEL	14.23	YES	NO	2.52	8.4	NO	YES
AKAP6	Binds to the regulatory subunit of protein kinase A, highly expressed in brain and cardiac tissue	WT	WLIDMESLV	7.29	YES	YES	4.14	-	NO	NO
		E/K	WLIDMKSLV	13.55	YES	YES	4.20	12.3	NO	NO

^1^ CD8 response was evaluated by multimers, and subsequent reactivity was measured by the expression of CD107 A/B and IFN-y of the tetramer-positive clones against tumor cells in vitro [30].

**Table 4 vaccines-11-01174-t004:** Top 10 atomic interactions between ASTN1 and HLA-A*02:01. Highlighted in yellow are the interactions differentially expressed in the neoantigen and not in its wild-type counterpart. The nomenclature of the atoms corresponds to that used in the CHARMM36 force field.

	ASTN1 Wild-Type			ASTN1 Neoantigen		
#	Contact Type	Mean Score	Interaction (HLA–Peptide)	Contact Type	Mean Score	Interaction (HLA–Peptide)
1	H-bond	0.977	Resid. 147, atom NE1—Resid. 9, atom O	H-bond	0.988	Resid. 147, atom NE1—Resid. 9, atom O
2	Other	0.961	Resid. 146, atom NZ—Resid. 10, atom C	H-bond	0.965	Resid. 146, atom NZ—Resid. 10, atom OT2
3	H-bond	0.96	Resid. 146, atom NZ—Resid. 10, atom OT1	H-bond	0.965	Resid. 159, atom OH—Resid. 1, atom O
4	H-bond	0.946	Resid. 146, atom NZ—Resid. 10, atom OT2	Other	0.96	Resid. 146, atom NZ—Resid. 10, atom C
5	Other	0.781	Resid. 143, atom OG1—Resid. 10, atom C	H-bond	0.958	Resid. 146, atom NZ—Resid. 10, atom OT1
6	Other	0.778	Resid. 146, atom CE—Resid. 10, atom OT1	H-bond	0.944	Resid. 99, atom OH—Resid. 3, atom N
7	Other	0.759	Resid. 147, atom CD1—Resid. 9, atom O	H-bond	0.891	Resid. 63, atom OE2—Resid. 1, atom N
8	Other	0.734	Resid. 146, atom CE—Resid. 10, atom OT2	H-bond	0.881	Resid. 63, atom OE1—Resid. 1, atom N
9	Other	0.715	Resid. 147, atom NE1—Resid. 8, atom CB	H-bond	0.877	Resid. 70, atom NE2—Resid. 7, atom NE1
10	Other	0.694	Resid. 146, atom CE—Resid. 10, atom C	Other	0.871	Resid. 63, atom CD—Resid. 1, atom N

## Data Availability

Docking and molecular dynamics simulation data of AKAP6 and ASTN1 peptides are available at Zenodo at https://doi.org/10.5281/zenodo.5772726 (created on 10 December 2021).

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
