# Peer review of "Usefulness of Docking and Molecular Dynamics in Selecting Tumor Neoantigens to Design Personalized Cancer Vaccines: A Proof of Concept"

_vaccines, 2023, doi:10.3390/vaccines11071174_

Round 1

Reviewer 1 Report

The manuscript explored docking and molecular dynamics to study the stability and immunogenicity of peptide-HLA complexes. It is an interesting study, however, the authors need to emphasise the significant findings or novelty of the work in the manuscript. Also, analysis with only 2 peptides may not be sufficient to draw a significant conclusion. The manuscript also lacks in vitro/in vivo assessment to justify the findings.  

Page 2, line 52, 54, 58 etc: Please italicise the in vivo throughout the manuscript.

Page 4, line 125-137: What are VMD and NAMD? Please write in full prior to the abbreviation.

Page 6, Table 3: Please specify how the in vitro data was obtained.

Page 10, Figure 3: Any statistical analysis? 

Author Response

Point 1. The manuscript explored docking and molecular dynamics to study the stability and immunogenicity of peptide-HLA complexes. It is an interesting study; however, the authors need to emphasise the significant findings or novelty of the work in the manuscript. Also, analysis with only 2 peptides may not be sufficient to draw a significant conclusion. The manuscript also lacks in vitro/in vivo assessment to justify the findings.

Response 1: This article was presented as a proof of concept to study the molecular and atomic characteristics of the immunogenicity of neoantigens in four peptides previously assessed in vitro. It is important to noted that these four peptides were selected from a total of 42 structures analyzed by molecular docking to perform the molecular dynamic simulations. Currently, we are carrying out molecular dynamics simulations with other peptides, but these results are outside the scope of this work and will be the subject of another publication.

In this study in vitro or in vivo experiments are not carried out because the peptides used for the docking and molecular simulations were selected from a list of neoantigens predicted and evaluated in vitro in the article “Targeting of cancer neoantigens with donor-derived T cell receptor repertoires” by Stronen et al. The in vitro experiments to evaluate the immunogenicity of the predicted neoantigens consist in the identification of neoantigen-specific CD8 T cells by multimers to sort and expand them. These expanded neoantigen-specific CD8 T cells were then co-cultured in vitro with autologous tumor cells from the patient to assess recognition via CD107a/b and IFN-y expression.

Point 2: Page 2, line 52, 54, 58 etc: Please italicise the in vivo throughout the manuscript.

Response 2: All terms in vivo, in vitro and in silico were italicised.

Point 3: Page 4, line 125-137: What are VMD and NAMD? Please write in full prior to the abbreviation.

Response 3: The abbreviation VMD means Visual Molecular Dynamics and the abbreviation NAMD means Nanoscale Molecular Dynamics. These are fully written prior to the abbreviation in line 129 in the non-tracked version of the document.

Point 4. Page 6, Table 3: Please specify how the in vitro data was obtained.

Response 4: The in vitro data was obtained by Strønen et al. through in vitro assays evaluating CD8 T cells response by multimers, and subsequent reactivity measured by expression of CD107 A/B and IFN-y of tetramer-positive clones against tumor cells in vitro as described in the foot note in table 1 and 3.

Point 5. Page 10, Figure 3: Any statistical analysis

Response 5: Unfortunately, it is not possible to perform statistical analysis to assess whether there is any statistically significant difference between distributions because in several cases there are less than 10 samples per distribution.

Reviewer 2 Report

In this manuscript, Amaya-Ramirez et al. conduct a proof-of-concept study on the use of molecular simulation techniques—namely, docking and molecular dynamics tools—to better identify tumor neoantigens that form stable HLA-A*02-peptide interactions for recognition by cytotoxic CD8+ T lymphocytes. Although monoclonal antibody-based immunotherapies for cancer treatment are now widely used, the development of cancer vaccines is very challenging and has received less attention. This study addresses a very significant challenge in cancer vaccine development which is the identification of immunogenic tumor neoantigens. The work is important to advance this type of cancer immunotherapy and worthy of publication. My comments (detailed below) are minor.

(1)    Abstract (line 23): “N-terminal” should be revised to “N-terminus”.

(2)    Introduction:  Revise the sentence on line 50 to read, “These in silico strategies allow prioritization of potentially immunogenic neoantigens”.

(3)    Table 1:  The description for Table 1 on lines 94-97 is redundant. The second sentence simply repeats the table title and should be removed. The Table 1 title should be revised to read, “Peptide sequences of the wild-type and mutant neoantigen selected for assessment using molecular docking”. The last sentence can be revised as follows: “Amino acid substitutions in the mutant neoantigens are presented in red”.

(4)    What are the functions, if known, of the genes associated with AKAP6 and ASTN1? Authors might want to include gene functions in Table 1.

(5)    Table 2 is not cited in text.  I suggest citing Table 2 after “…number of total interactions between the peptide and HLA” in line 105.

(6)    Why the difference in immunogenicity for peptide HELLS in Table 2 compared to Table 1? Does the immunogenicity need to be repeated in Table 2?

(7)    Table 4:  Please change “Peptido” in the category row to “Peptide”.

Author Response

Point 1. Abstract (line 23): “N-terminal” should be revised to “N-terminus”.

Response 1: We change the “N-terminal” for “N-terminus” in the abstract and main text.

Point 2. Introduction: Revise the sentence on line 50 to read, “These in silico strategies allow prioritization of potentially immunogenic neoantigens”.

Response 2: We change the line 50 as suggested.

Point 3. Table 1:  The description for Table 1 on lines 94-97 is redundant. The second sentence simply repeats the table title and should be removed. The Table 1 title should be revised to read, “Peptide sequences of the wild-type and mutant neoantigen selected for assessment using molecular docking”. The last sentence can be revised as follows: “Amino acid substitutions in the mutant neoantigens are presented in red”.

Response 3: We change the description of the table as suggested.

Point 4. What are the functions, if known, of the genes associated with AKAP6 and ASTN1? Authors might want to include gene functions in Table 1.

Response 4:

Considering the aim of the study, we find it relevant to describe only the function of the genes encoding the ASTN1 and AKAP6 peptides, to which docking and molecular dynamics simulations were performed. The functions of these genes were described in Table 3.

Point 5. Table 2 is not cited in text.  I suggest citing Table 2 after “…number of total interactions between the peptide and HLA” in line 105.

Response 5: We cited the table 2 in the text as suggested in line 106-107 in the non-tracked version.

Point 6. Why the difference in immunogenicity for peptide HELLS in Table 2 compared to Table 1? Does the immunogenicity need to be repeated in Table 2?

Response 6: The difference in immunogenicity for peptide HELLS between tables was due to an error, the correct information is in table 1. Additionally, the entire immunogenicity column of the table 1 was checked to verify the data. The immunogenicity column of table 2 was removed and it was mentioned in the text that the immunogenicity information was reported in table 1 on line 107 in the non-tracked version of the article.

Point 7. Table 4: Please change “Peptido” in the category row to “Peptide”.

Response 7: We change the header of table 4 to Peptide.

Round 2

Reviewer 1 Report

Thank you to the authors for their response. I recommend accepting the manuscript.